# A Comprehensive Evaluation of Tool-Assisted Generation Strategies

**Alon Jacovi**[1*]   **Avi Caciularu**[2]   **Jonathan Herzig**[2]

**Roee Aharoni**[2]   **Bernd Bohnet**[3]   **Mor Geva**[3]

[1]Bar Ilan University   [2]Google Research   [3]Google DeepMind
alonjacovi@gmail.com

## Abstract

A growing area of research investigates augmenting language models with tools (e.g., search engines, calculators) to overcome their shortcomings (e.g., missing or incorrect knowledge, incorrect logical inferences). Various few-shot tool-usage strategies have been proposed. However, there is no systematic and fair comparison across different strategies, or between these strategies and strong baselines that do not leverage tools. We conduct an extensive empirical analysis, finding that (1) across various datasets, example difficulty levels, and models, strong no-tool baselines are competitive to tool-assisted strategies, implying that effectively using tools with in-context demonstrations is a difficult unsolved problem; (2) for knowledge-retrieval tasks, strategies that *refine* incorrect outputs with tools outperform strategies that retrieve relevant information *ahead of* or *during generation*; (3) tool-assisted strategies are expensive in the number of tokens they require to work—incurring additional costs by orders of magnitude—which does not translate into significant improvement in performance. Overall, our findings suggest that few-shot tool integration is still an open challenge, emphasizing the need for comprehensive evaluations of future strategies to accurately assess their *benefits* and *costs*.

## 1 Introduction

Augmenting language models (LMs) with tools has been proposed to overcome LMs' inherent weaknesses (Mialon et al., 2023; Qian et al., 2022), such as the lack of grounding to reliable or updated sources (Jiang et al., 2023), incoherent logical ability (Liu et al., 2022; Ling et al., 2023) and arithmetic ability (Gao et al., 2023b), among others. This is done through *tool-assisted (TA) generation*, where LMs are trained or instructed to use external tools, such as search engines over the web—e.g.,

Google search (Gao et al., 2023a; Press et al., 2023; Nakano et al., 2022), Wikipedia search (Trivedi et al., 2022a), a calculator (Schick et al., 2023), or a python interpreter (Paranjape et al., 2023). Often, tool invocations are structured as *Chain-of-Thought* (CoT) long-form answers (Wei et al., 2023).

Recent work proposed a variety of strategies for interfacing between the LM and the tool, such as through demonstrations of API calls (Paranjape et al., 2023) or using the tool to refine the model's output (Gao et al., 2023a)—see Figure 2 for an overview. But what are the advantages and trade-offs of different TA strategies? For example, some strategies incur significantly higher *computation costs* than others with little to no improvement in performance. There is a gap in the literature on the *evaluation* of such strategies, in particular *against strong baselines* and *against each other*. Concretely, works that report empirical evaluations are often restricted to comparisons of a single proposed strategy against a limited selection of non-TA baselines, using a limited selection of LMs or even a single LM, or focus on evaluating various LMs with a specific TA strategy (Li et al., 2023). Additionally, comparisons often do not consider the increase in computation that each TA strategy requires, which vary significantly, and have a large effect on inference time or cost.

The above issues are only some of the pitfalls we observed in the literature, limiting the scope of current evaluations. In §3, we analyze the literature for common pitfalls and collect a set of guidelines towards a fair and reliable evaluation procedure specifically for TA strategies. Next (§4), we conduct a study which addresses all of the observed pitfalls, using GPT3, Flan-UL2 and Flan-PaLM, and complex reasoning benchmarks StrategyQA, MuSiQue, GSM8K, and DROP. We report a fair, systematic comparison of five few-shot TA strategies across multiple models and demonstrations, and all strategies use the same set of tools.

---

*Work done during an internship at Google Research.

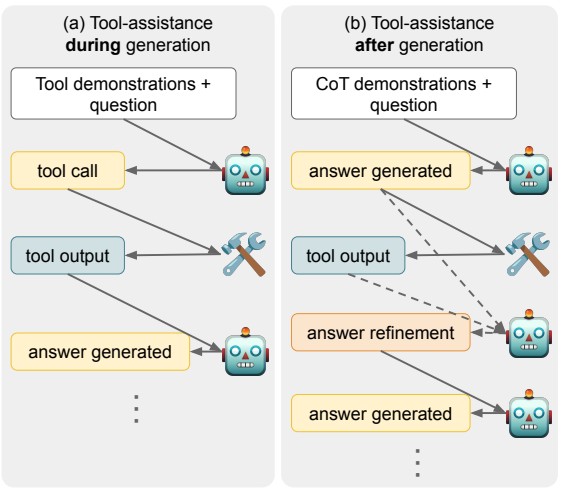

Figure 1: Illustration of tool-assistance strategies that invoke tools and insert their outputs into the prompt (a), and strategies that first generate some output, and only use tools to fix and refine it (b).

We analyze the study results (§5) and arrive at surprising conclusions: (1) Non-TA baselines are stronger than initially reported. In most cases, TA strategies do not significantly or at all improve on non-TA strategies on popular Question Answering datasets. (2) For retrieval tools in knowledge tasks, TA strategies that fix model output after it is generated perform better than TA strategies that prompt the model to interface with the tool directly during generation. For calculator tools in calculation-intensive tasks, the relationship is not decisive. (3) TA strategies incur significantly higher computation costs than non-TA baselines by multiplicative factors, and there is no general correlation between computation cost and performance, with the exception that refinement strategies in retrieval settings are more costly than non-refinement strategies.

In §6 we report a fine-grained analysis of the results. We investigate the effect of each example's difficulty—e.g., very large numbers, or very rare entities) on improvement from tool usage, and find that tools do not systematically improve model performance on harder examples, where they were expected to have the strongest improvement. Finally, based on an error analysis of failure cases, we find that the majority of mistakes follow incorrect tool invocations, rather than incorrect tool responses (in the case of the retrieval tool) or incorrect inferences based on correct tool usage.

In conclusion, we conduct an extensive evaluation of few-shot TA strategies, finding that previous estimates of tool-usage performance is not representative. Overall, this suggests that few-shot tool

integration is still an open challenge. We call the community to evaluate future strategies systematically, while taking into account the significant costs that these strategies require in comparison to their benefits. Towards this, we provide a set of concrete guidelines for fair and reliable evaluation of TA strategies. Moreover, We release the handcrafted collection of 184 demonstrations used in our study (attached in the supplementary material).

## 2 Tool-Assisted Language Models

We describe existing few-shot strategies for augmenting LMs with tools and discuss related work.

### 2.1 Few-shot TA strategies

Strategies for tool usage can be broadly divided into two categories: (a) Using tools during generation and insert the tools' outputs into the model's prompt (Figures 1a, 2a); (b) Using tools to refine the LM's output after generation (Figures 1b, 2b). Strategies can be further categorized into settings where the tool is heuristically called in a pipeline or called when the model generates pre-specified tool calls. Refer to Mialon et al. (2023) for a review of the literature on TA strategies and models.

Among TA strategies of type (a): **SelfAsk** (Press et al., 2023) decomposes the task into subtasks as simpler questions, such that a tool can be called on each question. A related strategy is *Demonstrate-Search-Predict* (Khattab et al., 2023). **Inline** strategies such as Toolformer (Schick et al., 2023)[1], ART (Paranjape et al., 2023), inter alia (Chen et al., 2022; Gao et al., 2023b; Lyu et al., 2023) demonstrate tool usage with pre-defined words or tokens and tool arguments, halt generation when those tokens and arguments are generated, invoke the tool, and insert its output into the prompt to resume generation. **Interleaving Retrieval** (Trivedi et al., 2022a) does not directly instruct the model to use tools, but calls the tool on each reasoning step, to provide the model with additional context for future steps. (Jiang et al., 2023) propose a similar strategy, opting to re-write each step after using it as a query. There are also strategies such as **Decomposed Prompting** (Khot et al., 2023) that are generalizations of the previous strategies.

Among TA strategies of type (b): **RARR** (Gao et al., 2023a) involves a pipeline designed for knowledge-based tasks: verifying the relevance

---

[1] Schick et al. primarily discusses tool usage with training. We adapt only the few-shot strategy in our experiments.

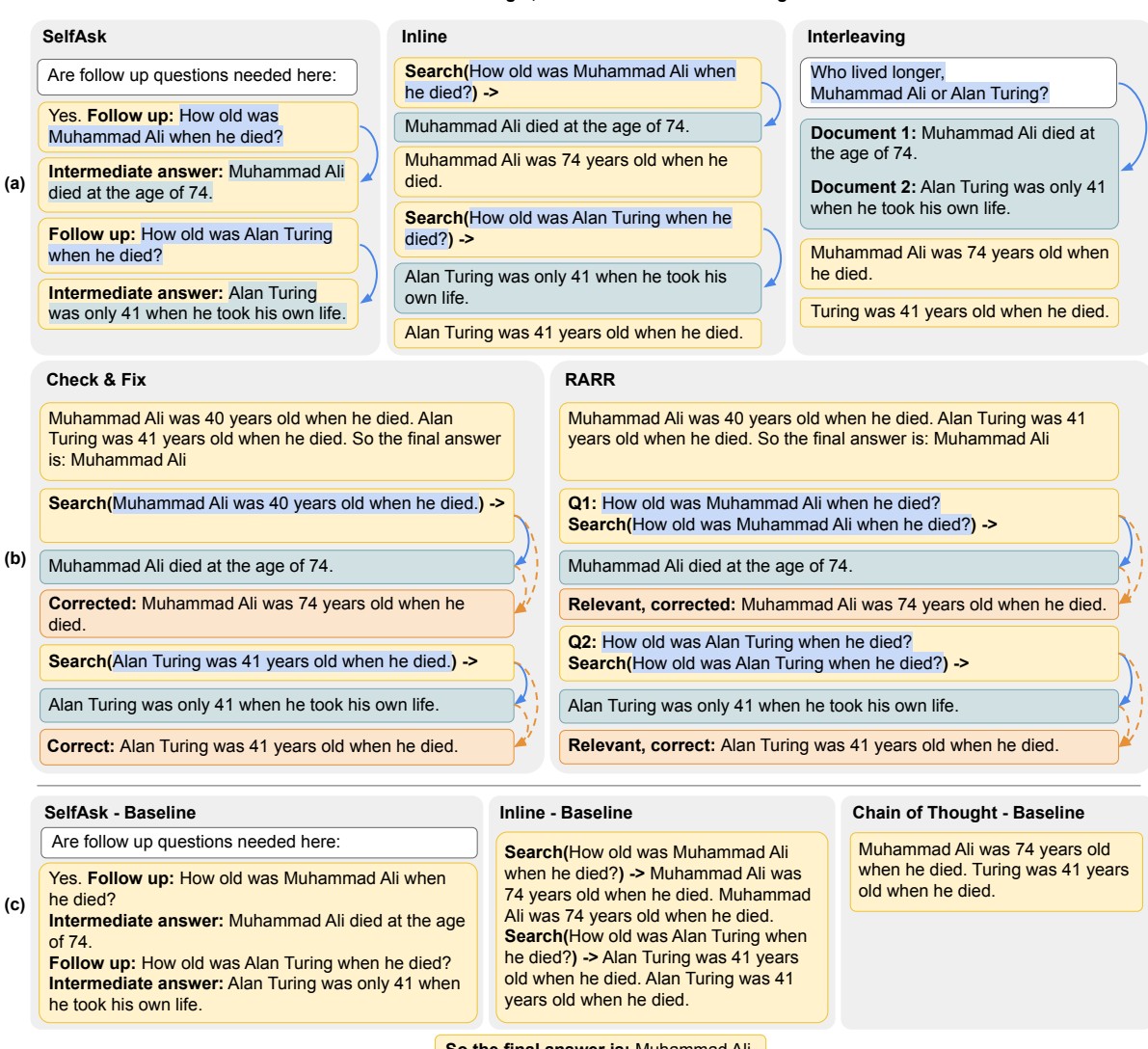

Figure 2: Overview of the TA strategies implemented in this work. Blue text marks tool queries, tool responses are in turquoise cells, refinement is in orange cells and dashed arrows, and yellow cells are LM generations.

and factuality of each claim by generating questions based on the claim, retrieving snippets that answer these questions, and checking if the answers match the information in the claim. If not, the claim is refined to match the snippets. **Check & Fix**, a method we introduce in this work, uses each CoT step as a search query, and checks whether the step is entailed by the retrieved snippets by prompting the model to classify this entailment. This strategy is similar to Jiang et al. (2023, contemporaneous work), which additionally uses low-confidence filtering but omits the entailment verification.

## 2.2 Related Work

**Training LMs to use tools.** While we are primarily concerned with few-shot tool assistance of LM generation, the literature also explores LMs which

are trained to use specific tools (Parisi et al., 2022; Hao et al., 2023; Patil et al., 2023). These methods are constrained to the tools seen during training, and require data (annotated, bootstrapped, or synthetically constructed) of tool demonstrations.

**Other tool-assisted neural networks.** There is adjacent research on augmenting neural networks, in ways besides textual interfaces, with tools (e.g., Andor et al., 2019; Jacovi et al., 2019) or training differentiable subnetworks that heavily mimic tools (Neelakantan et al., 2017; Trask et al., 2018).

## 3 Evaluation Pitfalls

While there is a plethora of TA strategies (§2.1), no systematic comparison of these strategies has been conducted. Research that proposes TA strategies in

| | Pitfall | Recommendation |
|---|---|---|
| **(1)** | Coupling the TA strategy and the tool together. | Comparisons of TA strategies should use the same tools across strategies. |
| **(2)** | Forcing no-tool baselines to the framework of the TA strategy. | The optimal way to solve the task without tools may be different from solving the task with tools: No-tool baselines should include multiple variants of both free-form and structured strategies, to ensure the TA strategies are not given an advantage. |
| **(3)** | Using one model across all comparisons. | Different models may behave differently when it comes to using tools effectively, based on their training data. Multiple models should be tested, if possible. |
| **(4)** | Using one prompt and set of demonstrations across all comparisons. | Multiple different sets of demonstrations should be used to get reliable estimates of few-shot performance. |
| **(5)** | Not considering TA strategy costs. | TA strategies can be efficient or inefficient with regards to the prompt tokens and generation tokens they require to work, with respect to no-tool baselines or with respect to each other. The differences can be significant (§5). Comparisons of TA strategies should factor the computation cost of the strategy, which we term as *token efficiency*. |

Table 1: Summary of evaluation pitfalls of TA strategies (§3) and recommendations to mitigate them.

few-shot settings is often not focused on evaluating properties of those strategies, but other aspects of LM capabilities (Press et al., 2023; Gao et al., 2023a), usage in particular strict contexts (Paranjape et al., 2023), evaluating various LM models themselves with a particular strategy (Mialon et al., 2023), and so on.

Below we collect observations from the literature that demonstrate the limited evaluation scope of TA strategies, in an effort to establish a set of criteria for future evaluations to be reliable and fair (a summary is provided in Table 1).

**(1) Coupling the TA strategy and the tool together.** Comparisons may vary the tools and methods together (e.g., a TA strategy *A* with a tool *A* versus a TA strategy *B* with a tool *B*).

**(2) Forcing baselines to the framework of the TA strategy.** Typical baselines to a given TA strategy are to apply that strategy while letting the model generate the tool's output instead of the tool, and using CoT prompting. However, the optimal way to solve the problem without tools may not be the same as the TA strategy in question. In this work, we implement three different baselines (§4) and find that there is *no clear winner* among two of them (we explore this empirically in §5).

**(3) Using one model across all comparisons.** Often, a single model is chosen to use as the underlying model for the TA strategy. This limits the insights from the evaluation to this model in particular, since conclusions may not carry over to other models. In this work, we find that the *best-performing strategies vary significantly* across different LMs (we explore this empirically in §5).

**(4) Using one prompt and one set of demonstrations across all comparisons.** Few-shot evaluation is known to be unreliable when using a single set of demonstrations as a single prompt (Perez et al., 2021). Furthermore, some prompts used in TA strategy evaluations—in particular, CoT demonstrations—appear so often on the internet that they are suspected to be part of the models' training data, further compromising their function (Jacovi et al., 2023).

**(5) Not considering TA strategy costs.** In many cases, the TA strategy requires significantly more compute than no-tool baselines, and different TA strategies also require different amounts of computation. Computation cost is not traditionally considered in comparisons.

# 4 Experimental Setup

Our goal is to conduct a fair and reliable comparison of *TA strategies*, without being influenced by properties of specific models, tools or prompts. To this end, we focus on *few-shot* tool usage, a popular TA scheme that allows flexibility around using new tools and adapting tools to specific tasks.

In what follows, we describe our experimental setup. What guides this experimental setup is to perform a comprehensive, rigorous evaluation without the pitfalls of §3. Our evaluation covers 5 different TA strategies, 4 recent LMs, 4 complex reasoning datasets, 3 few-shot prompts, and 2 tools. For each *TA strategy + dataset + model* combination, we run three experiments with a different number of demonstrations. Overall, our evaluation includes an execution of 342 experiments, each of which

generates 250 (GPT-3) or 500 (non-GPT-3) long-form answers. Additional implementation details are in Appendix A.

**Tool-assisted strategies.** We evaluate the TA strategies shown in Figure 2: SelfAsk, Inline, Interleaving, C&F and RARR. We additionally include variants of SelfAsk and Inline where the model is separately called to summarize tool output in relevant context, as it can often be very long (SelfAskQA and InlineQA; see Appendix A for details). Finally, in the retrieval settings, we use Top-1 retrieval for all models, and additionally Top-5 retrieval for the Flan-PaLM-540B model (see "*Models*" below) to check whether additional retrieved information can improve performance despite the significantly longer input and processing cost.

For SelfAsk and RARR we use the original implementation provided by the methods' creators. We implement Interleaving (Trivedi et al., 2022a), as at the time of this research no implementation was available. Importantly, this implementation yields similar performance to that of existing approaches that combine CoT with retrieval from Wikipedia by He et al. (2022); Jiang et al. (2023) (see full results in Appendix B). Additionally, Jiang et al. (2023, Figure 4) implemented methods that apply retrieval and refinement over generated CoT that are similar to C&F and achieve similar performance to ours, as well (see Appendix B). For Inline, we are not aware of reports on few-shot performance of a similar strategy in the literature.

**Baseline strategies.** We use no-tool versions of SelfAsk, Inline, and standard CoT prompting. The SelfAsk and Inline baselines simply involve giving the model the prompts used for the tool-based versions, while disabling tool calls (such that the model generates the output in-place of the tools). These are the baselines used by Press et al. (2023) and Schick et al. (2023) respectively.

**Datasets.** We consider tasks that require complex reasoning, where models could potentially benefit from external tool usage. Specifically, we use StrategyQA (Geva et al., 2021) and MuSiQue (Trivedi et al., 2022b), which require reasoning about entity knowledge, and GSM8k (Cobbe et al., 2021) and DROP (Dua et al., 2019) that evaluate arithmetic reasoning. In DROP we select examples that have numerical answers. We randomly sample 500 examples from the development set of each dataset (with the exception of StrategyQA, whose test set has 229 examples), and use it for performance evaluation of UL2, Flan-PaLM-540B and Flan-PaLM-62B . For GPT-3, we use a subset of 250 examples of that set, due to cost. We use standard evaluation measures for every dataset (F1 in the case of MuSiQue). We provide data examples in Appendix A.

**Models.** We evaluate the methods across four LMs: Flan-UL2-20B (Tay et al., 2023), GPT-3 (`text-davinci-003`) (Brown et al., 2020), Flan-PaLM-540B and Flan-PaLM-62B (Chung et al., 2022). We omit GPT-3 experiments on RARR and Interleaving due to cost. Importantly, our focus is *not* in comparing performance of these models, but to use them as samples of different model instances and training schemes against which to compare different TA strategies.

**Tools.** We strictly use the same tools across all strategies, to ensure a fair comparison: Google Search (Press et al., 2023; Schick et al., 2023; Lewis et al., 2021) for knowledge tasks, and a calculator (Schick et al., 2023; Qin et al., 2023) for the calculation tasks. RARR, SelfAsk and Interleaving are designed for retrieval settings only, while Inline and Check & Fix can be used in all settings. For the retrieval settings using Google Search and Flan-PaLM-540B , we test retrieval with both the top 1 and top 5 tool-retrieved snippets: The two formats are designed to cover both cases where a shorter tool output may prevent the model's answer from degenerating, and a longer tool output may help the model with more relevant information.

**Few-shot demonstrations.** In order to overcome bias from using demonstrations from prior work that were likely seen during training (Jacovi et al., 2023), we re-annotate prompts for all TA strategies, datasets and tools. We randomly sample 8 examples from each dataset's training set, and annotate each example with demonstrations for each TA strategy. Some of the strategies call the model multiple times with different prompts (e.g., Check & Fix, RARR), which requires separate annotations. This effort results in a total of 184 annotated demonstrations, which we release as a resource for future works on TA generation. From each set of 8 demonstrations, we then construct three separate prompts—3-shot, 5-shot and 7-shot—randomly sampled from the original 8 demonstrations, to get a better estimation of few-shot performance.

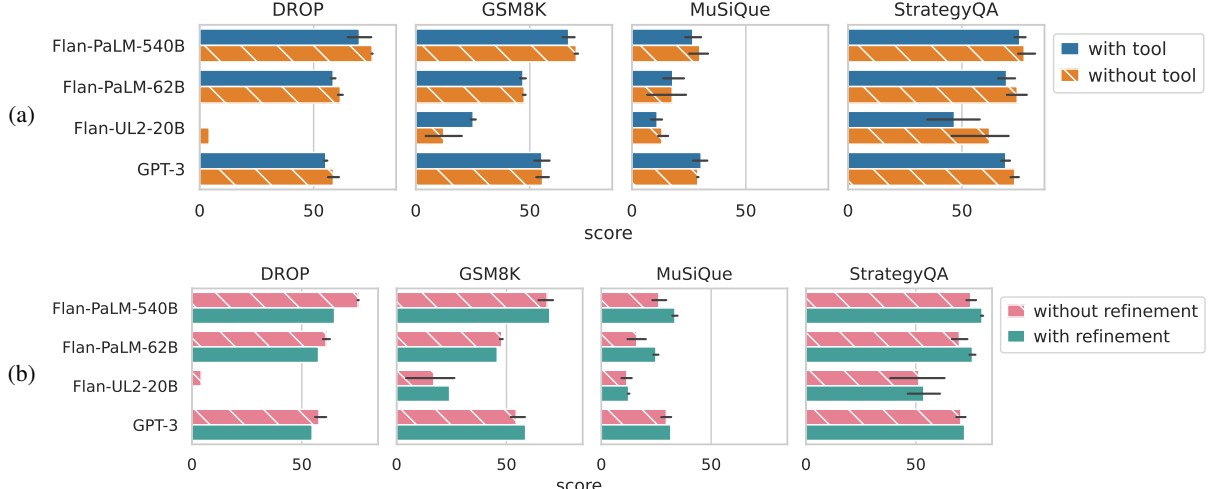

Figure 3: A comparison of evaluation scores across two areas (§5): (a) No-tool baselines vs. TA strategies; (b) Tool usage via refinement of generated text vs. tool usage during generation, where the generated text contains tool arguments is conditioned on tool outputs. The dark line marks the confidence interval among samples.

## 5 Comparative Results

**Organization of the results.** Due to the

**Tool vs. no tool.** Previous work that propose TA strategies found that using such strategies consistently improve performance in comparison to no-tool baselines (Press et al., 2023; Jiang et al., 2023; Trivedi et al., 2022a, inter alia).

Figure 3 shows that the TA strategies do not improve performance over the no-tool baselines in our selection of datasets. The figure shows results against the average of the different few-shot scores, though we observe similar trends when using the maximum of scores as well. Full results are in Appendix B. Similarly to us, Gao et al. (2023a, §6.2) found that StrategyQA performance slightly decreased with tools in RARR compared to no-tool baselines for PaLM-540B (Chowdhery et al., 2022), and Jiang et al. (2023, §6.2) found that performance decreased on StrategyQA in two settings comparable to our implementations of Interleaving and Check & Fix with GPT-3.

We conclude that for the settings in this work, *the no-tool baselines are stronger than initially expected based on the literature.* More research is required to investigate whether this relationship holds in other contexts, though we note that the datasets and models used in our experiments are common in TA research (Mialon et al., 2023).

Additionally, our experiments provide empirical justification to Recommendations (2) and (3) in §3. First, we find that the CoT and Inline baselines outperform each other at a roughly equal rate, and neither emerges as a clear winner. This shows

| Model | Dataset | Best strategy |
|---|---|---|
| GPT-3 | StrategyQA | Baseline-Inline |
| GPT-3 | DROP | Baseline-Inline |
| GPT-3 | GSM8K | Check & Fix |
| GPT-3 | MuSiQue | Inline |
| Flan-PaLM-540B | StrategyQA | Baseline-CoT |
| Flan-PaLM-540B | DROP | Baseline-Inline |
| Flan-PaLM-540B | GSM8K | Baseline-Inline |
| Flan-PaLM-540B | MuSiQue | RARR-Top5 |
| Flan-UL2-20B | StrategyQA | Baseline-Inline |
| Flan-UL2-20B | DROP | Baseline-Inline |
| Flan-UL2-20B | GSM8K | Inline |
| Flan-UL2-20B | MuSiQue | Baseline-CoT |
| Flan-PaLM-62B | StrategyQA | Baseline-CoT |
| Flan-PaLM-62B | DROP | Baseline-CoT |
| Flan-PaLM-62B | GSM8K | Inline |
| Flan-PaLM-62B | MuSiQue | Check & Fix |

Table 2: For each combination of dataset and model, we derive the best-performing strategy on the average score across the few-shot prompts. Notably, *the best-performing strategy varies across different models, datasets or prompts*, which means that it is necessary to evaluate over all axes to get a better estimation of general performance.

that different baselines obtain different results, and so, relying on only a single baseline in evaluation does not necessarily provide a good estimation for no-tool performance (recommendation (2)). Also, *the best-performing strategies vary significantly across models*, which highlights the importance of using multiple models for evaluation (recommendation (3))—for illustration, we report the highest-performing strategies in each setting in Table 2, to

| TA strategy | Prompt tokens (*canonical*) | Prompt tokens (*empirical*) | | | |
|---|---|---|---|---|---|
| | | Retrieval | | Calculator | |
| | | GPT-3 | Flan-PaLM-540B | GPT-3 | Flan-PaLM-540B |
| Baseline | $n$ | 353 | 353 | 1418 | 801 |
| SelfAsk | $t(n+\frac{kt+1}{2})$ | 2281 | 1399 | - | - |
| SelfAskQA | $t(2n+k)$ | 3589 | 2736 | - | - |
| Inline | $t(n+\frac{kt+1}{2})$ | 1793 | 1775 | 3453 | 1083 |
| InlineQA | $t(2n+k)$ | 3375 | 3672 | - | - |
| Check & fix | $t(2n+k)$ | 3839 | 3547 | 7548 | 3647 |
| RARR | $3n(t+1)$ | | 4729 | - | - |
| Interleaving | $t(n+\frac{kt+1}{2})$ | | 3221 | - | - |

Table 3: Average number of prompt tokens per strategy (5-shot), with $n$ as the CoT prompt length, $t$ as the number of tool calls, $k$ as the tool's output length. Flan-PaLM-540B has a shorter context window than GPT-3, which limits prompt length. The canonical formula for RARR favorably assumes a single verification question.

| TA strategy | Answer tokens (*canonical*) | Answer tokens (*empirical*) | | | |
|---|---|---|---|---|---|
| | | Retrieval | | Calculator | |
| | | GPT-3 | Flan-PaLM-540B | GPT-3 | Flan-PaLM-540B |
| Baseline | $m$ | 44 | 42 | 58 | 88 |
| SelfAsk | $m$ | 20 | 72 | - | - |
| SelfAskQA | $2m$ | 59 | 64 | - | - |
| Inline | $m$ | 103 | 248 | 62 | 102 |
| InlineQA | $2m$ | 114 | 256 | - | - |
| Check & fix | $2m$ | 89 | 177 | 75 | 177 |
| RARR | $3m$ | | 181 | - | - |
| Interleaving | $m$ | | 72 | - | - |

Table 4: Average number of answer tokens across the 5-shot experiments, for each strategy. The RARR formula assumes a single verification question per step.

show that the overall conclusion can be distorted by choosing a particular model or strtegy Extended details are in Appendix B.1.

**Tool use during generation vs. post-generation refinement.** In Figure 3 we compare the strategies that use tools during generation against the strategies that first generate an answer, and then use tools to improve the answer. For retrieval tasks, refinement clearly outperforms non-refinement strategies, but the same does not apply to the calculation tasks. We conjecture that planning calculations ahead of time during generation is more aligned with LM pretraining data, based on internet text, than planning retrieval queries in similar contexts.

**Token efficiency.** TA strategies are typically evaluated in terms of task performance and properties such as factuality and logic correctness. We argue that computational cost is another important factor to consider. Specifically, we propose to evaluate *token efficiency*, that is, the amount of *prompt* tokens

and *generated* tokens, which have direct effect on the cost of the TA strategy. Notably, the cost of a TA strategy depends on various variables, including model size, GPU type, caching optimizations, vocabulary size, beam search size, and so on. However, token counts can serve as a plausibly generic proxy for the purpose of comparing the cost of different TA strategies, as other factors are roughly equal across strategies, as long as the same models and tools are used. We consider prompt tokens and generated tokens separately, as they often have different consequences on cost.[2]

Tables 3, 4 show both canonical and empirical comparisons across TA strategies with regards to token efficiency. The canonical comparison is a function of the relevant variables in the "canonical" setting where the model was expected to answer

---

[2]Depending on model architecture and quantity of times reusing the same prompt, prompt processing cost can be optimized, whereas the token generation cost varies with other factors such as vocabulary size.

the question perfectly, and use the tool perfectly as intended. Across all TA strategy experiments, we found *no general correlation* between token efficiency and performance. Concretely: (1) All TA strategies are significantly more expensive than the no-tool baselines by orders of magnitude, while not incurring an improvement worthy of this extra cost. *Empirically, using tools in each case can incur extra costs by a factor of 5x to 10x for prompt processing, and 2x to 5x for generation.* (2) The refinement strategies are more expensive than the no-refinement strategies. So while they improve performance for retrieval tasks, it comes at a cost.

# 6 Analytical Results

We discuss further analyses of our results, findings that (a) our observations generally hold across different levels of example difficulty, and (b) most prediction errors of tool-augmented LMs stem from incorrect inputs to the tool and bad outputs from it, and not from a lack of tool usage.

## 6.1 Example Difficulty

It has been shown that LMs have difficulty solving problems involving long-tail entities (Kandpal et al., 2022; Mallen et al., 2022) and complex mathematical reasoning challenges (Mishra et al., 2022; Imani et al., 2023). Accordingly, we ablate the results from §5 along the following axes of example difficulty, in order to understand how tools can affect performance on difficult examples. We provide an overview of the trends here, and extended results are available in Appendix B.

**Measures of difficulty.** We investigate the effectiveness of tool-usage across varying levels of example difficulty, which we approximate in two axes: *(A) Long-tail entities (retrieval):* Following Mallen et al. (2022), we extract the entities from the question and associated gold answers in StrategyQA and MuSiQue, and use the corresponding entity Wikipedia page views as a measure of popularity. *(B) Large numbers (calculation):* We segment the examples in the calculation tasks based on the range of the median and largest number in the example (question and gold solution in GSM8k, or question and context paragraph in DROP).

**Results.** Performance across increasing levels of entity popularity and computation complexity, with different LMs and TA strategies, are shown in Figure 4a and Figure 4b, respectively. We find that performance uniformly decreases for harder ex-

amples in the retrieval setting for all models, but in the calculation setting, this only manifests for Flan-UL2-20B (implying that the larger models are more robust to the numerical ranges in GSM8K and DROP). Overall, in all cases *tool use does not improve upon the baselines even when controlling for the harder cases where tools are expected to be more useful*. This conclusion is aligned with our error analysis in §6.3, which shows that the common errors stem from incorrect tool arguments, more than correct tool arguments but incorrect inferences based on them. Flan-UL2 with a calculator is an exception, where tool use indeed helps, though moreso on the *easier* examples, likely due to a higher rate of correct arguments to the calculator.

## 6.2 Tool Usage Statistics

A possible explanation for the similar performance of no-tool baselines could be a lack of tool usage. To check this, we aggregate usage over the different TA strategies, and find that the models indeed use tools in the majority of the cases; 70%-80% in SelfAsk, and >90% in others (see Appendix B). We also investigate usage across other axes, such as models and number of demonstrations, and find similar trends. However, the datasets and tasks we investigate are designed to benefit from the tools in all cases, which shows that few-shot demonstrations are not always sufficient in inducing tool use in models. In particular, the SelfAsk strategies receive the lowest tool use, being the strategies that use natural language to query whether to use the tool (the answer begins with "Are follow up questions needed here:' to which the model answers "No" in the cases where the tool is not used).

## 6.3 Error Analysis

We sampled 50 instances for which an error was made by the TA models, randomly across the 5-shot experiments, and categorized them across three categories: (A) Incorrect tool input; (B) incorrect tool output; (C) incorrect model inferences based on correct tool usage. Error B applies only to the retrieval settings, where the retrieval tool (Google Search in our case) retrieved a wrong or irrelevant snippet. The errors were distributed approximately to 60% (A), 10% (B), and 30% (C) in the retrieval setting, and 80% (A) and 20% (C) in the calculation setting. Li et al. (2023) reported an error analysis for tool-assistance in dialogue customer assistance settings, with similar conclusions regarding error A, although errors B and C do not apply in their

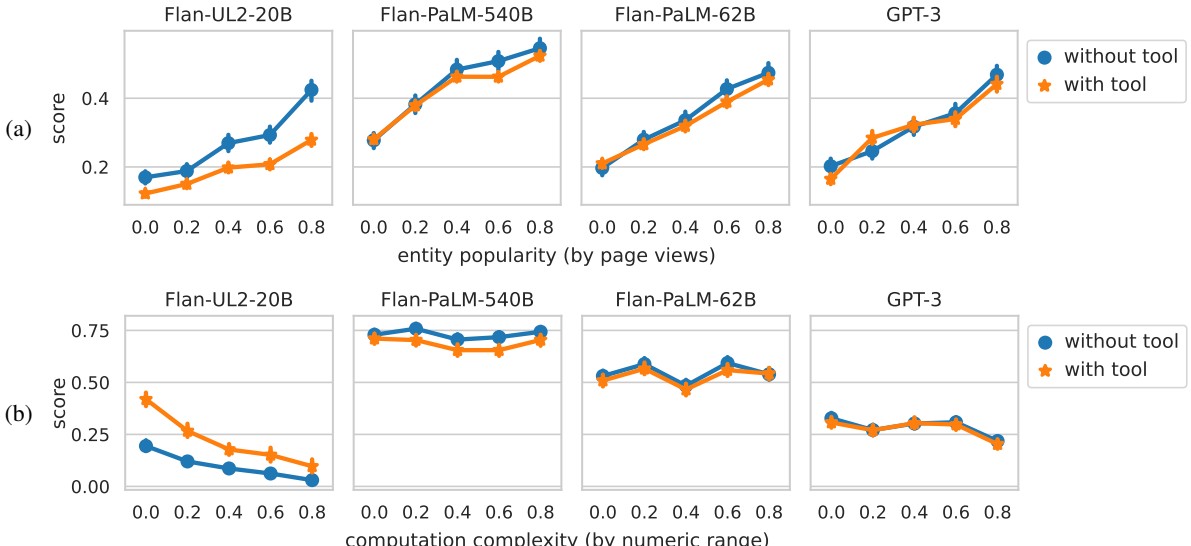

Figure 4: We analyze performance of the strategies across two area (no-tool baselines vs. TA strategies), conditioned on *example difficulty* as defined by the existence of rare or common entities in the retrieval settings (via percentile of page views) and small or large numbers in the calculation settings (via percentile of numeric range). In (a), lower page views imply higher difficulty, and in (b), larger numbers imply higher difficulty.

context, and other error types manifest instead.

Our results suggest that the majority of errors are not due to the incorrect tool responses (i.e., issues with Google Search as a choice of retriever), and overall more influenced by incorrectly invoking tools to begin with, in comparison to invoking them correctly but composing the solution incorrectly.

## 7 Conclusions and Takeaways

We conduct a comprehensive assessment of few-shot tool augmentation strategies for LMs, covering hundreds of experiments with multiple LMs, datasets, and tools. Our experiments show that current tool-usage integration approaches are presently a false promise; prompting strategies that do not use tools typically obtain similar task performance, without the high cost of tool execution. Controlling for example difficulty, where tools are expected to provide the most benefit, does not explain the relative strength of the no-tool baselines. Instead, the primary errors we observe are related to incorrect usage of the tools to begin with (i.e., generating incorrect arguments to the tool).

Our findings call for more robust evaluation of future TA strategies, primarily in more practical settings where models are not expected to leverage inherent abilities to solve tasks. To this end, our work provides concrete evaluation guidelines, such as employing stronger baselines and factoring in computation costs.

## Limitations

While our study aims to provide a comprehensive evaluation of TA strategies, there are some limitations. First, recent work (Dodge et al., 2021; Magar and Schwartz, 2022; OpenAI, 2023) suggests that examples from public datasets, like those used in our evaluation, may have leaked to the training data of recent LMs. Such contamination can introduce biases to the evaluation, such as lack of need for external tools. We are not aware of alternatives without this issue at the time of this writing.

Second, due to the high cost of executing large LMs in an exhaustive evaluation, we ran only a single experiment for each combination of TA strategy, model, dataset, and number of demonstrations. However, given the sensitivity of models to the demonstrations (Perez et al., 2021), future work should extend this evaluation to use multiple sets of demonstrations for each such combination.

Last, while our findings show that non-tool models often perform on par with existing TA strategies, our setting favors tool usage. For example, our tasks only require a single type of tool such that the model does not need to choose between multiple tools. Future work that investigates when and how tools *can* improve performance should consider more realistic evaluation settings, for example, by considering tasks where the model may need to use multiple types of tools together, or tasks where tools may sometimes give unhelpful answers.

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

# A Implementation Details

## A.1 Tool-Assisted Strategies.

**General Details.** In all cases, if the tool invocation fails (e.g., with an ill-formatted calculation, or a null response from Google Search), the model is used to generate the tool's output instead. For all retrieval settings using Google Search, we test both Top-1 and Top-5 retrieval: The two formats are designed to cover both cases where a shorter tool output may prevent the model's answer from degenerating, and a longer tool output may help the model with more relevant information. Illustrative examples of the data are available in Table 5.

**SelfAsk and SelfAskQA.** SelfAsk involves decomposing each question into a series of simpler sub-questions, and calling the tool directly for each sub-question. The tool's output is inserted into the prompt as an intermediate answer. When the model generates a step that begins with the string "So the answer is:", it is expected to generate an answer that builds on the previous intermediate answers which were tool outputs. In this work, we use Google Search as the tool as in the original work by (Press et al., 2023).

Our SelfAsk implementation reuses the original implementation by Press et al. (2023). Since Self-Ask is designed specifically for knowledge-based QA, we only evaluate this strategy for the knowledge tasks MuSiQue and StrategyQA.

The SelfAskQA variant involves calling the model for each pair of sub-question and retrieved snippet that (hopefully) contains its answer. This method of recursively calling the model with different a different prompt as if it were another tool is a technique proposed by Khot et al. (2023). We collect all sub-questions from the SelfAsk prompts in order to construct QA prompts (using the tool to retrieve supporting snippets). The model is called with the QA prompts in order to answer each sub-question based on its snippet. The SelfAskQA variant in essence summarizes each Google Search snippet, which can be as long as a paragraph, into a short answer to the given sub-question, effectively simplifying and shortening the overall answer.

Among the two SelfAsk implementations, neither decisively outperforms the other: SelfAskQA outperforms SelfAsk for GPT-3 and Flan-PaLM-62B on both MuSiQue and StrategyQA, but for Flan-PaLM-540B and Flan-UL2-20B the relationship flips.

**Inline and InlineQA.** The Inline strategy format largely mimics the Toolformer format by Schick et al. (2023), but can also be cast into the ART framework by Paranjape et al. (2023) or the Decomposed Prompting framework by Khot et al. (2023). In general, the strategy simply calls for generating the tool call in a predefined format—in our case, square brackets and the tool name. The tool is invoked with the arguments generated by the model inside the brackets, and the tool's output is inserted into the model. Our implementation is based on the inference code implemented by Schick et al. (2023), although notably, we focus on few-shot usage, and do not perform the tool-usage pretraining step that largely concerns the referenced work.

We implement two variants: Inline, which uses a tool called "Search" that appends the retrieved snippet or calculation output directly into the prompt, and InlineQA, which uses a tool called "QA" which calls the model with a separate prompt in order to summarize the retrieved snippet into a concise answer, identically to the aforementioned SelfAskQA variant. As with the SelfAsk and SelfAskQA variants, among Inline and InlineQA in the knowledge-based tasks, neither consistently outperforms the other in particular.

**Interleaving.** The Interleaving Retrieval strategy (Trivedi et al., 2022a) proposes to use each reasoning step by the model in its CoT answer as a query to a retrieval model. The retrieved snippet is then added to the prompt in order to provide additional information to the model. The structure for each demonstration becomes: (1) All retrieved documents thus far; (2) The question; (3) The generated answer thus far (see Trivedi et al., 2022a for details). In this way, the tool is used heuristically without explicit demonstrations from the model, but the generation of the answer at each CoT step is still conditioned on tool usage based on the previous steps.

**Check & Fix.** We propose this strategy as a more lightweight variant of refinement based on tools in comparison to RARR, and it is comparable to contemporaneously proposed (Jiang et al., 2023): After each CoT step, the step is checked for accuracy using a tool, and if found inaccurate, a new fixed step is generated to replace it.

In the *retrieval* setting, each step is verified and fixed by prompting the model to classify whether the step is contradicted by the retrieved paragraphs,

| Model | Dataset | Strategy | Example | Model Answer |
|---|---|---|---|---|
| GPT-3 | StrategyQA | Baseline-CoT | **Question**: Was a woman Prime Minister directly before or after Stanley Baldwin? **Answer**: No. | Stanley Baldwin was Prime Minister of the United Kingdom from 1923 to 1929. The woman Prime Minister directly before him was Margaret Thatcher, who served from 1979 to 1990. The woman Prime Minister directly after him was Theresa May, who served from 2016 to 2019. So the answer is no. |
| Flan-UL2 | MuSiQue | SelfAsk | **Question**: What administrative territorial entity contains the place where KPRM is licensed to broadcast to? **Answer**: Hubbard County. | Are follow up questions needed here: Yes. What is the place where KPRM is licensed to broadcast to? Intermediate answer: KPRM is licensed to broadcast to the city of Houston. Follow up: What administrative territorial entity contains the city of Houston? Intermediate answer: The city of Houston is in Texas state. Follow up: What is the administrative territorial entity that contains Texas state? Intermediate answer: Texas state is in the United States. So the final answer is: The United States |
| Flan-PaLM-62B | DROP | Check & Fix | **Context**: *[omitted paragraph about the 2000 census]* **Question**: How many households and families reside in the county according to the 2000 census? **Answer**: 26484 | 26484 households and 12068 families reside in the county. There were 40543 people, 15416 households, and 11068 families residing in the county. Thus, 26484 + 12068 = 38552. So the final answer is: 38552. |
| Flan-PaLM-540B | GSM8K | Inline | **Question**: How many girls are in the school if 40% of a school population is made up of 240 boys? **Answer**: 360 | If 40% of the school population is made up of 240 boys, then 1% is made up of [Calculator(240 / 40) -> 6.0] 6 boys. Therefore, the total school population is made up of [Calculator(6 * 100) -> 600] 600 students. If the school population is made up of 600 students, and 40% of them are boys, then the remaining 60% are girls, a total of [Calculator(60 / 100) * 600 -> 360] 360 girls. So the final answer is: 360. |

Table 5: Illustrative examples of various datasets, strategies and model outputs. The answers from the Interleaving, Check & Fix and RARR models are of the same format as the CoT baseline.

and if so, to generate the fixed step based on demonstrations. In the *calculation* setting, each step is first heuristically checked for whether it contains a calculation, and if so, the calculation is inserted into the calculator tool, and the model is prompted to verify whether the tool output is consistent with the calculation in the text. If this is incorrect, the model generates the fixed step. In both cases, the answer generation continues where the fixed step completely replaces the original incorrect step.

**RARR.** RARR (Retrofit Attribution using Research and Revision, Gao et al., 2023a) was proposed as a post processing method for refining any text, including LM chain-of-thought outputs. This is done via automatically finding attribution for each claim in the text, and post-editing the output to fix unsupported content while preserving the original output as much as possible. Our RARR implementation reuses the original implementation by Gao et al. (2023a).

The RARR process involves the following steps, with each considered as a separate tool:

1. *Question Generation*: First, they generate a series of questions that cover various aspects of a passage, referred to as passage x. The questions generated aim to verify and attribute information from the passage. This is done via prompting the LM with few-shot examples.

2. *Evidence Retrieval*: For each generated question, the Google Search tool is utilized to retrieve the top-$k$ passages that are related to the question. In this work, we evaluate both Top-1 and Top-5.

3. *Evidence Ranking*: The retrieved evidences are next ranked using a query-document relevance model scorer. Unlike the original RARR implementation (Gao et al., 2023a), which uses the GTR retrieval model (Ni et al., 2022), we instead implement the scorer via few-shot LM prompting, as suggested by the authors. The output of this stage is thus the top-1 ranked evidence.

4. *Agreement Phase*: Given a triplet of a text, question, and an evidence, this phase determines whether both the text and the question imply the same answer to the question. This is implemented via few-shot LM prompting using a chain-of-thought style prompt.

5. *Editing Phase*: If the previous Agreement Phase outputs disagreement between the text and the evidence, the (text, question, evidence) triplet is fed to a model that outputs a revised

| Model | Dataset | Best baseline |
|---|---|---|
| GPT-3 | StrategyQA | Inline |
| GPT-3 | DROP | Inline |
| GPT-3 | GSM8K | CoT |
| GPT-3 | MuSiQue | Inline |
| Flan-UL2-20B | StrategyQA | Inline |
| Flan-UL2-20B | DROP | Inline |
| Flan-UL2-20B | GSM8K | CoT |
| Flan-UL2-20B | MuSiQue | CoT |
| Flan-PaLM-540B | StrategyQA | CoT |
| Flan-PaLM-540B | DROP | Inline |
| Flan-PaLM-540B | GSM8K | Inline |
| Flan-PaLM-540B | MuSiQue | CoT |
| Flan-PaLM-62B | StrategyQA | CoT |
| Flan-PaLM-62B | DROP | CoT |
| Flan-PaLM-62B | GSM8K | Inline |
| Flan-PaLM-62B | MuSiQue | CoT |

Table 6: For each combination of dataset and model, we derive the best-performing baseline on the average score across the few-shot experiments. *There is no clear winner*: Two of the baselines achieve the best score in 50% of cases.

version of the text, considering the discrepancy between the previous text and the evidence. This is implemented via few-shot LM prompting using a similar chain-of-thought style prompt from the previous stage (see Gao et al., 2023a for the exact prompting template). The agreement and editing phases run iteratively until there are no needed revisions, detected in the Agreement Phase.

### A.2 Baselines

**Chain-of-Thought.** The CoT baseline is the standard baseline proposed by Wei et al. (2023) and implemented as a baseline by Press et al. (2023); Paranjape et al. (2023), inter alia. Often, the demonstrations used for this baseline are those originally published by Wei et al. (2023). In this work we annotate a new sample of examples with CoT answers for the purpose of a better estimation of CoT few-shot performance, and release our annotations.

**Self-Ask.** The Self-Ask baseline uses the Self-Ask tool demonstrations, but does not invoke the tool after each "Follow up:" call, and instead generates the entire answer. This is the original no-tool baseline in Press et al. (2023).

**Inline.** The Inline baseline uses the Inline tool demonstrations, but does not invoke the tool after

| Model | Usage (%) |
|---|---|
| Flan-PaLM-540B | 70.9 |
| Flan-PaLM-62B | 80.6 |
| Flan-UL2-20B | 82.6 |
| GPT-3 | 95.1 |

Table 7: Note that RARR and Interleaving are guaranteed to use tools so they are omitted.

| Strategy | Usage (%) |
|---|---|
| Check & Fix | 92.9 |
| SelfAsk | 80.4 |
| SelfAskQA | 72.8 |
| Inline | 99.9 |
| InlineQA | 96.1 |

Table 8: Overview of average rate of tool usage across experiments. Note that RARR and Interleaving are guaranteed to use tools.

each tool call, and instead generates the entire answer. This is the original no-tool baseline in Schick et al. (2023).

## B  Extended Results

We provide the full results for our experiments (described in §4) in §B.1, and further analysis of TA strategy performance and tool usage in §B.2.

### B.1  Full Experiment Results

Tables 9, 10 detail our experiment results. Tables 11, 12, 13, 14 detail average and max aggregations over the few-shot prompts. As mentioned, we sample 500 examples for Flan-PaLM-62B , Flan-PaLM-540B and Flan-UL2-20B experiments, and 250 for GPT-3 experiments, with the exception of StrategyQA whose test set has 229 examples.

For DROP and MuSiQue, we report the F1 measures using the evaluation scripts provided by Dua et al. (2019); Trivedi et al. (2022b) respectively. For GSM8K, we normalize the numerical answers and measure exact-match. For StrategyQA, we normalize the answers (for capitalization, prefix and suffix punctuation, and so on) and measure exact-match to "yes" and "no".

**Best-performing strategies and baselines in each setting.** In Tables 2, 6 we show the best-performing baseline and best-performing general strategy for each setting of model and dataset, among the average scores across the three few-shot

experiments. For strategies in general (Table 2), we see that the winning strategies vary significantly for different models, which supports Guideline (3) in Table 1.

The distribution among the baselines is split 50%-50% among CoT and Inline. When considering each few-shot experiment separately (i.e., not taking the average), the distribution is 60.0%, 37.5%, and 2% for *Baseline-CoT*, *Baseline-Inline* and *Baseline-SelfAsk* respectively for which baseline achieves the best-performing score. This supports Guideline (2) in Table 1.

## B.2 Analysis

**Example Difficulty.** Figures 5, 6 show extended results for the example difficulty analyses in §6. Here we consider the median of each difficulty metric—i.e., the difficulty across all entities or numbers in the example—rather than the minimum or maximum, as well as the ablation of refinement strategies against no-refinement strategies. We additionally checked for two alternative axes: operation complexity (addition and substraction as "easy" examples, and multiplication and division as "hard" examples) and popularity links rather than popularity views. The trends we observe in the main paper hold in all of these cases.

**Tool Usage.** Tables 7, 8 show aggregate tool usage percentages over multiple axes. Overall, few-shot demonstrations induce tool usage in the majority of cases, though not completely so (i.e., below 100%).

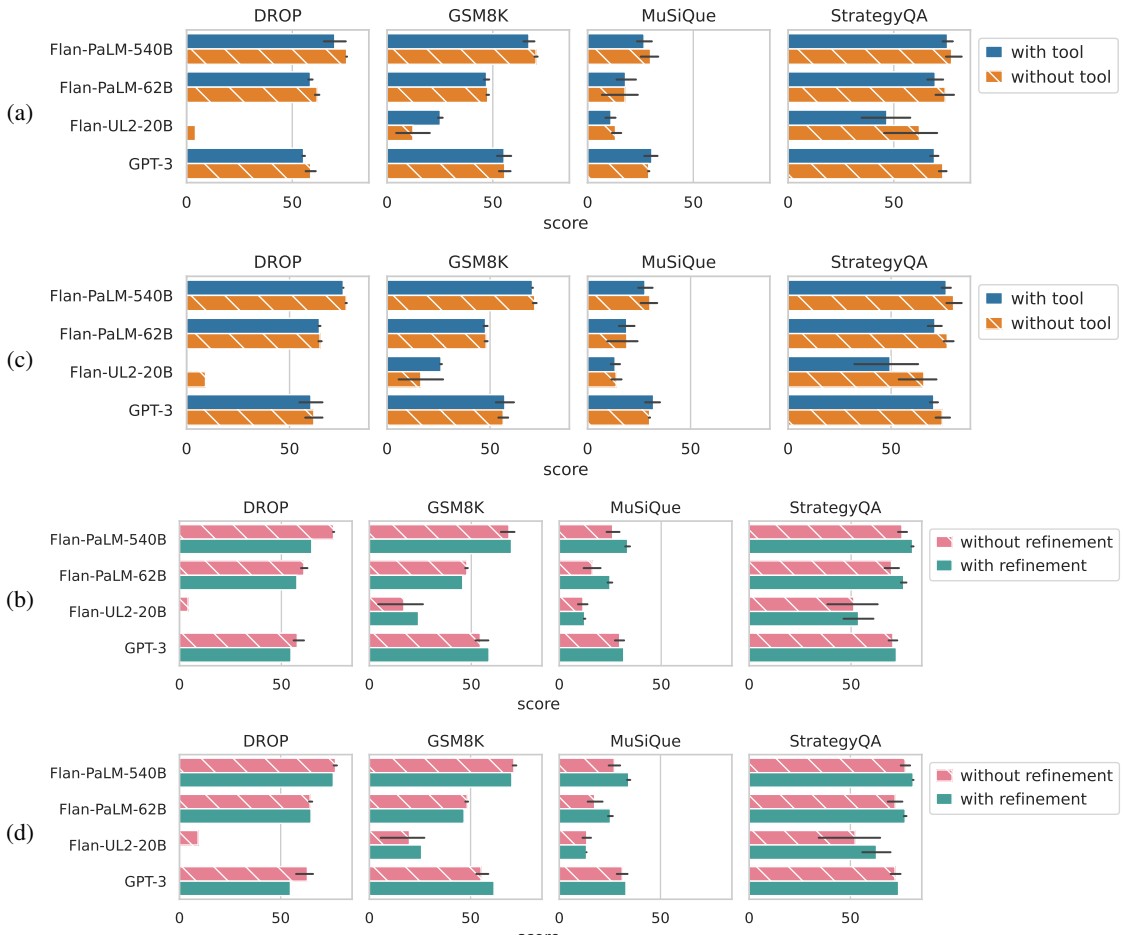

Figure 5: An extension of Table 3 with results for both the average across few-shot experiments (a-b) and the maximum across few-shot experiments (c-d)—i.e., the maximum between 3-shot, 5-shot and 7-shot for each experiments setting.

| Strategy | Model | MuSiQue | | | StrategyQA | | |
|---|---|---|---|---|---|---|---|
| | | 3-shot | 5-shot | 7-shot | 3-shot | 5-shot | 7-shot |
| RARR | Flan-PaLM-540B | 34.86 | 35.09 | 34.14 | 80.35 | 81.22 | 80.79 |
| RARR | Flan-UL2-20B | 13.40 | 12.01 | 12.98 | 55.90 | 40.17 | 42.79 |
| RARR | Flan-PaLM-62B | 23.60 | 23.42 | 24.07 | 75.98 | 77.73 | 77.73 |
| Baseline-CoT | Flan-PaLM-540B | 33.07 | 33.36 | 33.80 | 79.91 | 84.28 | 82.10 |
| Baseline-CoT | Flan-UL2-20B | 15.14 | 16.50 | 16.10 | 67.25 | 71.62 | 72.05 |
| Baseline-CoT | GPT-3 | 27.37 | 29.31 | 30.25 | 70.74 | 71.62 | 71.62 |
| Baseline-CoT | Flan-PaLM-62B | 23.60 | 23.42 | 24.27 | 75.98 | 79.04 | 80.35 |
| Baseline-SelfAsk | Flan-PaLM-540B | 25.80 | 25.34 | 24.31 | 76.86 | 73.36 | 75.55 |
| Baseline-SelfAsk | Flan-UL2-20B | 11.40 | 11.52 | 11.52 | 34.06 | 48.47 | 53.71 |
| Baseline-SelfAsk | GPT-3 | 27.98 | 28.13 | 29.80 | 72.05 | 74.24 | 73.36 |
| Baseline-SelfAsk | Flan-PaLM-62B | 5.28 | 9.52 | 5.43 | 58.95 | 75.98 | 74.24 |
| Baseline-Inline | Flan-PaLM-540B | 30.39 | 30.71 | 31.19 | 71.62 | 79.91 | 72.49 |
| Baseline-Inline | Flan-UL2-20B | 13.66 | 13.33 | 9.74 | 72.05 | 68.56 | 71.18 |
| Baseline-Inline | GPT-3 | 29.11 | 30.33 | 28.15 | 70.31 | 75.98 | 78.60 |
| Baseline-Inline | Flan-PaLM-62B | 23.42 | 22.69 | 21.86 | 75.11 | 73.36 | 75.55 |
| SelfAsk | Flan-PaLM-540B | 20.02 | 23.14 | 23.26 | 71.62 | 71.18 | 73.80 |
| SelfAsk | Flan-UL2-20B | 11.86 | 7.68 | 7.41 | 49.78 | 25.76 | 23.14 |
| SelfAsk | GPT-3 | 24.38 | 24.15 | 22.33 | 64.19 | 67.25 | 65.94 |
| SelfAsk | Flan-PaLM-62B | 13.79 | 14.80 | 12.68 | 67.25 | 67.69 | 66.38 |
| SelfAskQA | Flan-PaLM-540B | 21.08 | 21.92 | 22.91 | 71.62 | 69.43 | 73.80 |
| SelfAskQA | Flan-UL2-20B | 8.53 | 5.35 | 2.30 | 47.16 | 17.03 | 11.79 |
| SelfAskQA | GPT-3 | 32.74 | 31.30 | 30.34 | 65.50 | 67.69 | 70.31 |
| SelfAskQA | Flan-PaLM-62B | 15.42 | 17.49 | 14.51 | 67.25 | 68.12 | 69.00 |
| InlineQA | Flan-PaLM-540B | 31.86 | 32.78 | 32.10 | 70.31 | 72.93 | 73.36 |
| InlineQA | Flan-UL2-20B | 18.07 | 17.94 | 1.56 | 71.18 | 70.31 | 56.77 |
| InlineQA | GPT-3 | 34.90 | 36.65 | 31.32 | 70.31 | 72.05 | 70.31 |
| InlineQA | Flan-PaLM-62B | 12.52 | 11.65 | 10.55 | 61.14 | 63.32 | 61.57 |
| Check & Fix | Flan-PaLM-540B | 30.73 | 33.17 | 33.48 | 80.35 | 80.79 | 78.17 |
| Check & Fix | Flan-UL2-20B | 10.90 | 11.77 | 13.52 | 52.40 | 60.70 | 69.87 |
| Check & Fix | GPT-3 | 29.66 | 32.95 | 32.26 | 72.05 | 73.80 | 70.74 |
| Check & Fix | Flan-PaLM-62B | 25.21 | 26.39 | 26.47 | 75.55 | 71.18 | 76.42 |
| Inline | Flan-PaLM-540B | 18.97 | 24.42 | 22.61 | 74.24 | 74.24 | 75.11 |
| Inline | Flan-UL2-20B | 14.70 | 14.93 | 14.78 | 48.47 | 52.84 | 44.98 |
| Inline | GPT-3 | 28.85 | 31.03 | 33.54 | 70.31 | 69.43 | 68.56 |
| Inline | Flan-PaLM-62B | 9.95 | 9.45 | 13.32 | 54.59 | 68.56 | 70.31 |
| Interleaving | Flan-PaLM-540B | 23.71 | 21.29 | 20.51 | 76.86 | 78.60 | 75.98 |
| Interleaving | Flan-PaLM-62B | 23.43 | 23.71 | 24.42 | 74.67 | 71.62 | 74.24 |
| RARR-Top5 | Flan-PaLM-540B | 36.12 | 35.40 | 35.44 | 80.35 | 79.91 | 79.91 |
| SelfAskQA-Top5 | Flan-PaLM-540B | 19.75 | 21.60 | 21.99 | 69.87 | 70.31 | 72.05 |
| Inline-Top5 | Flan-PaLM-540B | 32.67 | 34.53 | 31.69 | 65.50 | 77.73 | 72.93 |
| Check & Fix-Top5 | Flan-PaLM-540B | 31.74 | 32.68 | 33.87 | 78.60 | 81.66 | 81.22 |

Table 9: Results for the knowledge-retrieval tasks of MuSiQue and StrategyQA. MuSiQue scores are F1 scores. Missing cells, such as "Interleaving" with Flan-UL2-20B, are experiments where the model failed to converge to an answer.

| Strategy | Model | DROP | | | GSM8K | | |
|---|---|---|---|---|---|---|---|
| | | 3-shot | 5-shot | 7-shot | 3-shot | 5-shot | 7-shot |
| Baseline-CoT | Flan-PaLM-540B | 77.2 | 75.0 | 74.2 | 67.4 | 70.8 | 70.8 |
| Baseline-CoT | Flan-UL2-20B | | | | 7.2 | 27.2 | 26.2 |
| Baseline-CoT | GPT-3 | 57.6 | 55.6 | 55.6 | 58.8 | 58.0 | 58.4 |
| Baseline-CoT | Flan-PaLM-62B | 65.6 | 63.6 | 59.2 | 47.4 | 46.2 | 47.4 |
| Baseline-Inline | Flan-PaLM-540B | 77.8 | 75.6 | 74.4 | 69.8 | 72.6 | 71.2 |
| Baseline-Inline | Flan-UL2-20B | | | | 3.6 | 5.6 | 3.6 |
| Baseline-Inline | GPT-3 | 57.6 | 66.0 | 59.6 | 51.6 | 54.0 | 53.2 |
| Baseline-Inline | Flan-PaLM-62B | 59.0 | 64.0 | 59.2 | 48.8 | 47.8 | 48.0 |
| Inline | Flan-PaLM-540B | 76.2 | 75.2 | 74.4 | 61.4 | 61.8 | 70.6 |
| Inline | Flan-UL2-20B | | | | 26.6 | 26.2 | 26.0 |
| Inline | GPT-3 | 56.8 | 66.0 | 45.2 | 50.8 | 52.4 | 52.8 |
| Inline | Flan-PaLM-62B | 57.0 | 64.0 | 57.8 | 48.8 | 47.8 | 48.2 |
| Check & Fix | Flan-PaLM-540B | 76.0 | 73.6 | 45.0 | 68.4 | 70.4 | 70.2 |
| Check & Fix | Flan-UL2-20B | | | | 23.2 | 25.8 | 23.2 |
| Check & Fix | GPT-3 | 54.8 | 54.4 | 54.8 | 56.0 | 58.4 | 61.6 |
| Check & Fix | Flan-PaLM-62B | 65.0 | 63.6 | 44.2 | 46.8 | 44.0 | 46.6 |

Table 10: Results for the calculator settings of DROP and GSM8K. We omit Flan-UL2-20B results on DROP, as the model could not converge to solve the task with our prompts, likely since each example in this task is very long.

| Strategy | Aggregation | Model | MuSiQue | StrategyQA |
|---|---|---|---|---|
| Baseline-CoT | Max | GPT-3 | 30.2 | 71.6 |
| Baseline-CoT | Average | GPT-3 | 29.0 | 71.3 |
| Baseline-CoT | Max | Flan-UL2-20B | 16.5 | 72.1 |
| Baseline-CoT | Average | Flan-UL2-20B | 15.9 | 70.3 |
| Baseline-CoT | Max | Flan-PaLM-62B | 24.3 | 80.3 |
| Baseline-CoT | Average | Flan-PaLM-62B | 23.8 | 78.5 |
| Baseline-CoT | Max | Flan-PaLM-540B | 33.8 | 84.3 |
| Baseline-CoT | Average | Flan-PaLM-540B | 33.4 | 82.1 |
| Baseline-SelfAsk | Max | GPT-3 | 29.8 | 74.2 |
| Baseline-SelfAsk | Average | GPT-3 | 28.6 | 73.2 |
| Baseline-SelfAsk | Max | Flan-UL2-20B | 11.5 | 53.7 |
| Baseline-SelfAsk | Average | Flan-UL2-20B | 11.5 | 45.4 |
| Baseline-SelfAsk | Max | Flan-PaLM-62B | 9.5 | 76.0 |
| Baseline-SelfAsk | Average | Flan-PaLM-62B | 6.7 | 69.7 |
| Baseline-SelfAsk | Max | Flan-PaLM-540B | 25.8 | 76.9 |
| Baseline-SelfAsk | Average | Flan-PaLM-540B | 25.1 | 75.3 |
| Baseline-Inline | Max | GPT-3 | 30.3 | 78.6 |
| Baseline-Inline | Average | GPT-3 | 29.2 | 75.0 |
| Baseline-Inline | Max | Flan-UL2-20B | 13.7 | 72.1 |
| Baseline-Inline | Average | Flan-UL2-20B | 12.2 | 70.6 |
| Baseline-Inline | Max | Flan-PaLM-62B | 23.4 | 75.5 |
| Baseline-Inline | Average | Flan-PaLM-62B | 22.7 | 74.7 |
| Baseline-Inline | Max | Flan-PaLM-540B | 31.2 | 79.9 |
| Baseline-Inline | Average | Flan-PaLM-540B | 30.8 | 74.7 |

Table 11: Aggregations by few-shot prompt of the results in Table 9 (baselines).

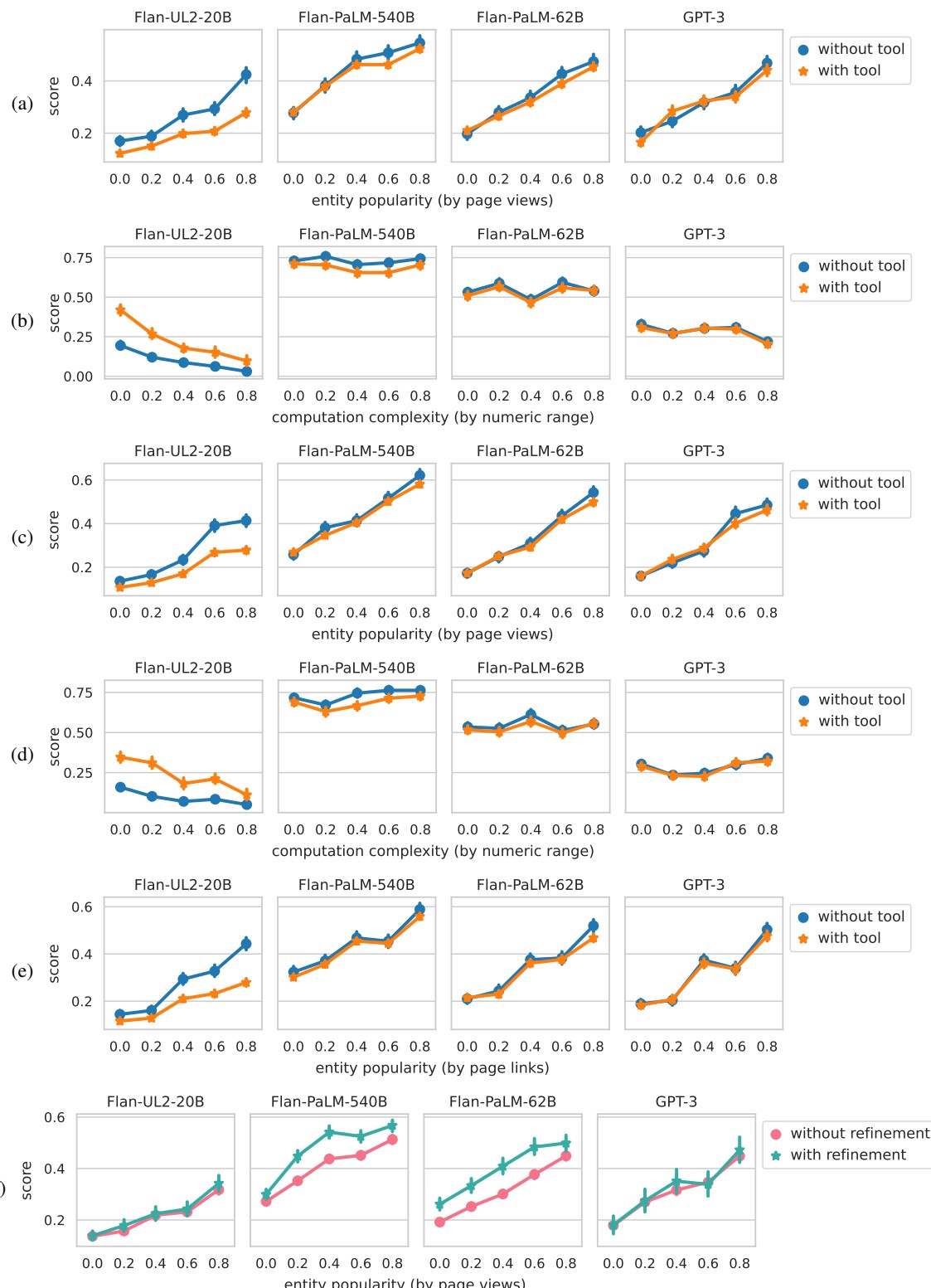

Figure 6: An extension of Table 4. (a-b) refer to taking the minimum of entity page views to ablate examples that have rare entities, and maximum of numbers to ablate examples with large numbers. (c-e) take the median in both cases, and (f) shows the results when comparing TA strategies between refinement and non-refinement types.

| Strategy | Aggregation | Model | MuSiQue | StrategyQA |
|---|---|---|---|---|
| Interleaving | Max | Flan-PaLM-62B | 24.4 | 74.7 |
| Interleaving | Average | Flan-PaLM-62B | 23.9 | 73.9 |
| Interleaving | Max | Flan-PaLM-540B | 23.7 | 78.2 |
| Interleaving | Average | Flan-PaLM-540B | 21.8 | 77.0 |
| RARR | Max | Flan-UL2-20B | 13.4 | 55.9 |
| RARR | Average | Flan-UL2-20B | 12.8 | 46.3 |
| RARR | Max | Flan-PaLM-62B | 24.1 | 77.7 |
| RARR | Average | Flan-PaLM-62B | 23.7 | 77.1 |
| RARR | Max | Flan-PaLM-540B | 35.1 | 81.2 |
| RARR | Average | Flan-PaLM-540B | 34.7 | 80.6 |
| RARR-Top5 | Max | Flan-PaLM-540B | 36.1 | 80.3 |
| RARR-Top5 | Average | Flan-PaLM-540B | 35.7 | 80.1 |
| Check & Fix | Max | GPT-3 | 32.9 | 73.8 |
| Check & Fix | Average | GPT-3 | 31.6 | 72.2 |
| Check & Fix | Max | Flan-UL2-20B | 13.5 | 69.9 |
| Check & Fix | Average | Flan-UL2-20B | 12.1 | 61.0 |
| Check & Fix | Max | Flan-PaLM-62B | 26.5 | 76.4 |
| Check & Fix | Average | Flan-PaLM-62B | 26.0 | 74.4 |
| Check & Fix | Max | Flan-PaLM-540B | 33.5 | 80.8 |
| Check & Fix | Average | Flan-PaLM-540B | 32.3 | 79.6 |
| Check & Fix-Top5 | Max | Flan-PaLM-540B | 33.9 | 81.7 |
| Check & Fix-Top5 | Average | Flan-PaLM-540B | 32.8 | 80.5 |

Table 12: Aggregations by few-shot prompt of the results in Table 9 (TA strategies).

| Strategy | Aggregation | Model | MuSiQue | StrategyQA |
|---|---|---|---|---|
| SelfAsk | Max | GPT-3 | 24.4 | 67.2 |
| SelfAsk | Average | GPT-3 | 23.6 | 65.8 |
| SelfAsk | Max | Flan-UL2-20B | 11.9 | 49.8 |
| SelfAsk | Average | Flan-UL2-20B | 9.0 | 32.9 |
| SelfAsk | Max | Flan-PaLM-62B | 14.8 | 67.7 |
| SelfAsk | Average | Flan-PaLM-62B | 13.8 | 67.1 |
| SelfAsk | Average | Flan-PaLM-540B | 22.3 | 72.2 |
| SelfAsk | Max | Flan-PaLM-540B | 23.4 | 74.2 |
| SelfAskQA | Max | GPT-3 | 32.7 | 70.3 |
| SelfAskQA | Average | GPT-3 | 31.5 | 67.8 |
| SelfAskQA | Max | Flan-UL2-20B | 8.5 | 47.2 |
| SelfAskQA | Average | Flan-UL2-20B | 5.4 | 25.3 |
| SelfAskQA | Max | Flan-PaLM-62B | 17.5 | 69.0 |
| SelfAskQA | Average | Flan-PaLM-62B | 15.8 | 68.1 |
| SelfAskQA | Max | Flan-PaLM-540B | 22.8 | 75.1 |
| SelfAskQA | Average | Flan-PaLM-540B | 21.9 | 71.9 |
| SelfAskQA-Top5 | Max | Flan-PaLM-540B | 22.0 | 72.1 |
| SelfAskQA-Top5 | Average | Flan-PaLM-540B | 21.1 | 70.7 |
| InlineQA | Max | GPT-3 | 36.7 | 72.1 |
| InlineQA | Average | GPT-3 | 34.3 | 70.9 |
| InlineQA | Max | Flan-UL2-20B | 18.1 | 71.2 |
| InlineQA | Average | Flan-UL2-20B | 12.5 | 66.1 |
| InlineQA | Max | Flan-PaLM-62B | 12.5 | 63.3 |
| InlineQA | Average | Flan-PaLM-62B | 11.6 | 62.0 |
| InlineQA | Max | Flan-PaLM-540B | 32.4 | 73.4 |
| InlineQA | Average | Flan-PaLM-540B | 32.1 | 72.2 |
| Inline | Max | GPT-3 | 33.5 | 70.3 |
| Inline | Average | GPT-3 | 31.1 | 69.4 |
| Inline | Max | Flan-UL2-20B | 14.9 | 52.8 |
| Inline | Average | Flan-UL2-20B | 14.8 | 48.8 |
| Inline | Max | Flan-PaLM-62B | 13.3 | 70.3 |
| Inline | Average | Flan-PaLM-62B | 10.9 | 64.5 |
| Inline | Max | Flan-PaLM-540B | 24.3 | 74.7 |
| Inline | Average | Flan-PaLM-540B | 22.0 | 74.2 |
| InlineQA-Top5 | Max | Flan-PaLM-540B | 34.5 | 77.7 |
| InlineQA-Top5 | Average | Flan-PaLM-540B | 33.0 | 72.1 |

Table 13: Aggregations by few-shot prompt of the results in Table 9 (TA strategies).

| Strategy | Aggregation | Model | DROP | GSM8K |
|---|---|---|---|---|
| Baseline-CoT | Max | GPT-3 | 57.6 | 58.8 |
| Baseline-CoT | Average | GPT-3 | 56.3 | 58.4 |
| Baseline-CoT | Max | Flan-UL2-20B | | 27.2 |
| Baseline-CoT | Average | Flan-UL2-20B | | 20.2 |
| Baseline-CoT | Max | Flan-PaLM-62B | 65.6 | 47.4 |
| Baseline-CoT | Average | Flan-PaLM-62B | 62.8 | 47.0 |
| Baseline-CoT | Max | Flan-PaLM-540B | 77.2 | 70.8 |
| Baseline-CoT | Average | Flan-PaLM-540B | 75.5 | 69.7 |
| Baseline-Inline | Max | GPT-3 | 66.0 | 54.0 |
| Baseline-Inline | Average | GPT-3 | 61.1 | 52.9 |
| Baseline-Inline | Max | Flan-UL2-20B | 9.2 | 5.6 |
| Baseline-Inline | Average | Flan-UL2-20B | 4.2 | 4.3 |
| Baseline-Inline | Max | Flan-PaLM-62B | 64.0 | 48.8 |
| Baseline-Inline | Average | Flan-PaLM-62B | 60.7 | 48.2 |
| Baseline-Inline | Max | Flan-PaLM-540B | 77.8 | 72.6 |
| Baseline-Inline | Average | Flan-PaLM-540B | 75.9 | 71.2 |
| Check & Fix | Max | GPT-3 | 54.8 | 61.6 |
| Check & Fix | Average | GPT-3 | 54.7 | 58.7 |
| Check & Fix | Max | Flan-UL2-20B | | 25.8 |
| Check & Fix | Average | Flan-UL2-20B | | 24.1 |
| Check & Fix | Max | Flan-PaLM-62B | 65.0 | 46.8 |
| Check & Fix | Average | Flan-PaLM-62B | 57.6 | 45.8 |
| Check & Fix | Max | Flan-PaLM-540B | 76.0 | 70.4 |
| Check & Fix | Average | Flan-PaLM-540B | 64.9 | 69.7 |
| Inline | Max | GPT-3 | 66.0 | 52.8 |
| Inline | Average | GPT-3 | 56.0 | 52.0 |
| Inline | Max | Flan-UL2-20B | | 26.6 |
| Inline | Average | Flan-UL2-20B | | 26.3 |
| Inline | Max | Flan-PaLM-62B | 64.0 | 48.8 |
| Inline | Average | Flan-PaLM-62B | 59.6 | 48.3 |
| Inline | Max | Flan-PaLM-540B | 76.2 | 70.8 |
| Inline | Average | Flan-PaLM-540B | 75.3 | 64.5 |

Table 14: Aggregations by few-shot prompt of the results in Table 10.