# OpenReview forum: "A Comprehensive Evaluation of Tool-Assisted Generation Strategies"
_EMNLP/2023/Conference — EMNLP 2023 Findings_

### Official Review · Reviewer_2Ti3 · 2023-07-21

**Typos Grammar Style And Presentation Improvements:** 1. Section 4 can be summarized and it…
**Soundness:** 3

**Excitement:**

3: Ambivalent: It has merits (e.g., it reports state-of-the-art results, the idea is nice), but there are key weaknesses (e.g., it describes incremental work), and it can significantly benefit from another round of revision. However, I won't object to accepting it if my co-reviewers champion it.

**Paper Topic And Main Contributions:**

This paper conducts a comprehensive evaluation of tool-assisted generation strategies. In specific, the authors first analyze the shortcomings of existing analysis and evaluation on tool-assisted strategies from five perspectives. To overcome these issues, the authors design a systematic evaluation scheme to evaluate five tool-assisted strategies across multiple models and complex reasoning benchmarks. The empirical analysis reveals several findings regarding the usage of tool-assisted strategies.

**Questions For The Authors:**

1. In Figure 3, the legend presents "w/" and "w/o" refinement (baselines), but the caption describes as "tool usage via refinement of generated text" vs "tool usage during generation". Which one is correct? For a more clear presentation, there seems to be three types of methods, baselines (w/o tool), tool-assisted during generation, tool-assisted refinement. Can they be put into the same figure?
2. Why does adopting calculator lead to a decrease of performance in DROP and GSM8K? It seems to be contradictory to some previous work. Is there any additional explanation or supportive analysis for this issue?
3. In the caption of Table 10, what do you mean "Flan-UL2-20B could not converge"? It is supposed to be few-shot in-context learning, why will the model converge without training?

**Reasons To Accept:**

1. This work investigates a timely and important problem in the era of LLMs.
2. The analysis of pitfalls in existing studies is valuable.
3. This work conducts a comprehensive and systematic evaluation with various tool-assisted strategies across multiple models and complex reasoning benchmarks.
4. The extensive analysis provides some new insights for the future studies on tool-assisted generation.

**Reasons To Reject:**

1. The organization of this paper can be further improved. There are some severe issues regarding the readability of this paper. The analysis of experimental results is a bit messy, where the illustration (figure/table) is hard to be interpreted. Despite many presented evaluation results, it is not sure whether it is useful for supporting the conclusion, since there is no related discussions or it is difficult to locate the discussions.
2. The types of investigated tools are limited (only two: Google search and a math calculator). It is questionable regarding the generality of the conclusion.
3. There are some interesting findings that are contradictory to the existing findings from other papers, but they lack of detailed explanations or supportive analysis for the results.
4. Although it is good to adopt various models and tool-assisted strategies for evaluation, the comparison is rather messy. From the results in the tables, it just list a bunch of results but it is difficult to make a clear comparision.

**Reproducibility:**

3: Could reproduce the results with some difficulty. The settings of parameters are underspecified or subjectively determined; the training/evaluation data are not widely available.

**Reviewer Confidence:**

3: Pretty sure, but there's a chance I missed something. Although I have a good feel for this area in general, I did not carefully check the paper's details, e.g., the math, experimental design, or novelty.

---

> ### Author Rebuttal · Authors · 2023-08-28
>
> We thank the reviewer for the thoughtful and very detailed review, and for the constructive suggestions!
>
> Overall, the review contains points regarding the general writing and organization of the paper (with one exception, point 2 in “reasons to reject”), all of which we will address in a manuscript update. Thanks for this valuable constructive feedback. We address these concerns below in detail, enumerated to the “reasons to reject” points:
>
> #### **(1) Organization and writing:**
> We appreciate the reviewer’s constructive suggestions, which we will address in the revised manuscript. Specifically, we will follow the reviewer’s recommendations in the “presentations improvements” section and add the following (a) a high-level summary of section 4, (b) making the references to the appendix more specific such that it is easy to find the relevant parts, (c) clarifying the reviewer’s questions 1 and 3 about the figures and tables (see our answers below), and (d)  improving the tables’ layouts in the appendix and splitting them to sub-sections.
> Please let us know if you have any further suggestions, thanks!
>
> #### **(2) Types of tools:**
> Please note that the scope of our experiment setting was prohibitively large due to requiring hundreds of different experiments in order to get sufficiently robust conclusions - we found that every aggregation axis in the “pitfalls” overview mattered. Adding one more type of tool would multiply the experiments by a multiplicative factor (requiring 150-200 more experiments, depending on the tool). This is unfortunately very costly and time-consuming to execute. However, we believe that this is not a compromise since our work is primarily concerned with showing sufficient evidence that this kind of rigorous evaluation is necessary, rather than comparing tool-usage performance of specific tools. In addition, we believe that there is also value in the particular choice of tools we made (i.e., search and a calculator), as these are arguably of the most widely used and studied tools.
>
> #### **(3) Contradictory findings:**
> Our findings do not actually contradict previous work in many cases - for example, we replicate results for SelfAsk when using the same model, prompt, datasets, and baselines as in that work (including their observation that SelfAsk wins over the baseline). Where possible, we use the same code written by the original authors, as well (we use our own implementation only when public code is not available). The contradicting conclusion comes from performing the experiments by aggregation across many models, settings and prompts, which serves to strengthen our discussion about the pitfalls in section 3. We will add this discussion to the paper.
>
> #### **(4) Results organization**
> Thank you for this valuable feedback. We prioritized visual analyses of the results, rather than presenting full tables due to lack of space. For the camera-ready version, we will incorporate aggregated results of the full table from the appendix into the main paper to provide a more organized and concrete overview of the results in the main paper.
>
> #### **Regarding the questions:**
>
> We answer the “questions to the authors” below in-depth, which primarily concern communication and writing. We will revise the paper to incorporate these suggestions and make all the information here clearer.
>
> Q1. “Tool usage during generation” (as opposed to tool usage after generation) refers to “w/o refinement”. We will clear this up in the paper. This is a binary categorization of TA strategies which we referred to in multiple ways, which caused this confusion. [Category 1] includes strategies like SelfAsk which invoke a tool before or during model generation, while [Category 2] refers to strategies like Check&Fix which first generate some answer without a tool, then invoke a tool to refine/revise the generated text. The goal of this categorization is to get meaningful general insights across multiple TA strategies, in the interest of giving more resilient conclusions.
>
> Q2. We see two possible explanations:
>
> (A) Previous research traditionally uses a single model (commonly GPT-3), a single dataset, a single prompt, and a single no-tool CoT baseline. We report empirical evidence in our work that such specific choices can give conclusions that do not generalize. For example, for Flan-UL2 and US on GSM8K specifically, “Inline” was the best performing strategy (tab. 7) winning over the baselines, and this does not contradict previous papers’ results. Our contradicting conclusions come from aggregating results over many settings. Note that in some previous work (e.g., Toolformer), improvements with tools were demonstrated with fine-tuned TA models, rather than few-shot TA.
>
> (B) For DROP and GSM8K, as well as many popular calculation tasks, it appears that newer and bigger models can already perform the calculations well without tools, so tools become unnecessary in those cases, and only bring further complications. We can see this in fig. 4 where the larger models do not lose performance even on calculations with large numbers (in GSM8K and DROP, these numbers are approx. in the thousands), while Flan-UL2 does.
>
> Q3. Here we meant that few-shot prompting with DROP demonstrations on Flan-UL2 does not manage to learn the task sufficiently (negligible performance). This is consistent with previous work, and likely due to the fact that DROP examples include long and difficult paragraphs in addition to questions about those paragraphs.
>
> #### **Regarding typos and grammar:**
> Thank you for the detailed comments. We will implement all of them with our best effort, including a reorganization of Appendix B and rewriting of captions (see our detailed response to the organization and writing concern above).

---

### Official Review · Reviewer_uECc · 2023-08-03

**Soundness:** 4

**Excitement:**

4: Strong: This paper deepens the understanding of some phenomenon or lowers the barriers to an existing research direction.

**Paper Topic And Main Contributions:**

The paper evaluates the existing tool augmentation strategies for LLMs. Existing work in the field lacks thorough evaluation of these strategies with strong baselines and with each other. This paper points out the pitfalls in the evaluation setups and conducts a comprehensive assessment of these augmentation strategies covering hundreds of experiments with multiple LMs, datasets, and tools.
Results show that current tool-usage integration approaches are not as good as they are portrayed: the improvement obtained by augmenting LLMs is low and not stable. Further, these tools incur extra costs by a factor of 5x to 10x for prompt processing, and 2x to
5x for generation.


**Reasons To Accept:**

1) The paper provides a comprehensive assessment of the existing LLM tool augmentation strategies,  which has not been explored before.   It exposes the pitfalls of the existing evaluation setups which can help the NLP community in designing better evaluation systems for LLMs.

2)  Along with documenting the pitfalls, the paper provides concrete evaluation guidelines and recommendations to mitigate the pitfalls, such as employing stronger baselines and factoring in computation costs.

3) The paper performs an impressive number of experiments (324 to be precise) using different LLMs, difficulty levels of examples, datasets, and tools. The experiment setups are also sound.

4) The paper contains a thorough literature review of the existing approaches. The paper is also well-written and easy to understand.



**Reasons To Reject:**

I do not have any reasons to reject the paper.

**Reproducibility:**

2: Would be hard pressed to reproduce the results. The contribution depends on data that are simply not available outside the author's institution or consortium; not enough details are provided.

**Reviewer Confidence:**

3: Pretty sure, but there's a chance I missed something. Although I have a good feel for this area in general, I did not carefully check the paper's details, e.g., the math, experimental design, or novelty.

---

> ### Author Rebuttal · Authors · 2023-08-28
>
> We thank the reviewer for the thoughtful and positive review! We greatly appreciate the reviewer's recognition of the  effort that went into the literature review of existing approaches, and the extensive experimental scope.

---

### Official Review · Reviewer_YEbL · 2023-08-04

**Typos Grammar Style And Presentation Improvements:** None
**Soundness:** 3

**Excitement:**

3: Ambivalent: It has merits (e.g., it reports state-of-the-art results, the idea is nice), but there are key weaknesses (e.g., it describes incremental work), and it can significantly benefit from another round of revision. However, I won't object to accepting it if my co-reviewers champion it.

**Missing References:**

None

**Paper Topic And Main Contributions:**

This paper presents a comprehensive evaluation of how different tool-assisted strategies influence the LLM performance. The contributions of this paper are as follows: firstly it summarizes the evaluation pitfalls of tool-assisted (TA) strategies as a set of criteria which can facilitate reliable and fair evaluation work in the future; secondly, it clearly demonstrates the experimental setups by grouping the models and datasets with TA strategy and distinguishing the during-generation and post-generation refinement for the tool usage. This work will serve as fundamentals for future evaluation of TA strategies.

**Questions For The Authors:**

Q1: how tool use during generation and post-generation refinement influence long-tail entities retrieval and large-numbers calculation respectively? Could you please give more examples to illustrate this question?

**Reasons To Accept:**

In this paper, the evaluation schema is set up by measurements and the some of the authors’ findings can be referenced for more robust evaluation of future TA strategies. In addition, the paper is logically written.

**Reasons To Reject:**

Measures of difficulty only consider the long-tail entities retrieval and the large numbers calculation, so it may not be comprehensive enough. Also, the definition of prompt processing cost and how it relates to token efficiency are not explained clearly in the section 5, so the argument “using tools in each case can incur extra costs by a factor of 5x to 10x for prompt processing, and 2x to 5x for generation” is not convincing.

**Reproducibility:**

4: Could mostly reproduce the results, but there may be some variation because of sample variance or minor variations in their interpretation of the protocol or method.

**Reviewer Confidence:**

3: Pretty sure, but there's a chance I missed something. Although I have a good feel for this area in general, I did not carefully check the paper's details, e.g., the math, experimental design, or novelty.

---

> ### Author Rebuttal · Authors · 2023-08-28
>
> We thank the reviewer for the thorough review and insightful comments! We appreciate the affirmation that our work will be fundamental to future evaluations, and the proposals for improvements to the paper.
>
> We address below the two concerns raised in the review:
>
> #### **(1) On the scope of the example difficulty analysis:**
> In the paper we analyzed the effectiveness of TA strategies across varying levels of example difficulty, where difficulty was measured using entity popularity and numerical range, following previous work [1, 2]. We ran two additional analyses:
> 1. Based on complexity of operations - between addition, division, multiplication and so on.
> 2. Based on entity rarity as an alternative measure of difficulty, via rarity of entity links in Wikipedia.
>
> In both cases, tools are expected to improve performance on the “harder” examples, but we did not observe statistically significant improvement in most settings, as with the analyses in the paper. We will add these results to the camera-ready version, and hope that this sufficiently increases the scope of this analysis. If there are any suggestions for other axes for testing performance across increasing difficulty levels, we are more than happy to analyze them!
>
> #### **(2) On prompt/generation processing costs:**
> Thank you for this comment. We agree that this discussion can be improved, and we will do so for the camera-ready version. Specifically, the token counting metrics (canonical formulas, and empirically 5x to 10x for prompts, 2x to 5x for generation) are directly equivalent to monetary payment per token charged by popular model APIs such as OpenAI, Google and MosaicML, which means that using TA strategies will increase costs by the observed factors based on the additional tokens. We will add more precise details and definitions here.
>
> When the model can be deployed locally, the token counts are not a direct measure for processing cost. This cost depends on model size, GPU type, optimizations such as caching prompt embeddings, vocabulary size, beam search size, self-consistency implementation, and so on. However, token counts can serve as a plausibly generic proxy for the purpose of comparing different TA strategies, since all other factors are roughly equal across strategies, as long as the same models and tools are used. Regardless, we will add these discussions and formalizations to the paper, so future evaluations can make better cost judgements.
>
> #### **Regarding Q1:**
> The appendix (e.g., fig. 6) has this analysis, which we can expand and mention in the main paper. There are not many conclusions we can make here, except that refinement’s improvement over non-refinement in retrieval settings does not appear to change significantly between easier or more difficult examples, and is different depending on the specific model.
>
> Thanks!
> Authors
>
> [1] Yuan et al., 2023, https://arxiv.org/abs/2304.02015
>
> [2] Mallen et al., 2023, https://arxiv.org/abs/2212.10511

---

### Meta-Review · Area_Chair_NYTx · 2023-09-25

**Recommendation:** 4

**Metareview:**

This paper investigates how tool-assisted strategies impact LLM performance. It provides comprehensive empirical evaluation comparing strategies that use tools and those that do not leverage any tools, using multiple LMs, benchmarks, and tools. The findings, particularly that the current tool usage approaches are not as good as they are portrayed, and that they also incur extra costs, are interesting, novel, timely, and can benefit the community for future research on tool-assisted efforts with LLMs.

Concerns include the organisation and readability of the paper, and the limited types of tools that are investigated. The author responses provide further insights on the latter that have the potential to increase the scope of the contributions of this paper.

---

### Decision · Program_Chairs · 2023-10-07

**Decision:**

Accept-Findings

**Comment:**

This paper investigates how tool-assisted strategies impact LLM performance. It provides comprehensive empirical evaluation comparing strategies that use tools and those that do not leverage any tools, using multiple LMs, benchmarks, and tools. The findings, particularly that the current tool usage approaches are not as good as they are portrayed, and that they also incur extra costs, are interesting, novel, timely, and can benefit the community for future research on tool-assisted efforts with LLMs.

Concerns include the organisation and readability of the paper, and the limited types of tools that are investigated. The author responses provide further insights on the latter that have the potential to increase the scope of the contributions of this paper.